# Deep learning models link local cellular features with whole-animal growth dynamics in zebrafish

Shang-Ru Yang[1],* , Megan Liaw[2],* , An-Chi Wei[1,3] , Chen-Hui Chen[2]

**Animal growth is driven by the collective actions of cells, which are reciprocally influenced in real-time by the animal's overall growth state. Whereas cell behavior and animal growth state are expected to be tightly coupled, it is not yet determined whether local cellular features at the micrometer scale might correlate with the body size of an animal at the macroscopic level. By inputting 722 skin cell images and corresponding size data for each zebrafish larva into machine learning models, we determined that the Vision Transformer (ViT) with a random cropping and voting strategy was able to achieve high predictive performance (F-score of 0.91). Remarkably, analyzing as few as 27 skin cells within a single image of 0.01 mm$^2$ was sufficient to predict the individual's overall size, ranging from 0.9 to 3.1 mm$^2$. Using a gradient-weighted class activation map (Grad-CAM), we further identified the cellular features influencing the model's decisions. These findings provide a proof-of-concept that macroscopic organismic information may be de-encrypted from a snapshot of only a few dozen cells using deep learning approaches.**

## Introduction

At a micrometer scale, animal growth may be visualized as the collective action of cells, each of which exhibits different behaviors orchestrated to facilitate the overall growth process. Individual cells may proliferate, migrate and/or change their size and shape because of spatiotemporal regulatory cues associated with the growth state of the animal. Importantly, changes in growth state are typically defined at a length scale several orders of magnitude higher than that of a cell diameter. Whereas cell behaviors and the growth state of an animal are assumed to be tightly coupled through intricate cross-talk mechanisms, it is not yet determined whether the microscopic information present at the cellular level directly reflects or may be correlated with the macroscopic growth state of an animal. Here, we sought to explore such connections across different length scales in a vertebrate model and to identify key cell-level features that enable prediction of whole-animal growth state information. Insights gained from this exploration may inspire advances in diagnostic methods and provide mechanistic insights into various developmental disorders and growth defects.

The size of zebrafish larvae during the post-embryonic growth period often shows high variability, which is accounted for by well-established systems for accurate staging or quantification (Parichy et al, 2009; McMenamin et al, 2016). One intriguing feature of zebrafish growth is that during a specific developmental period, the skin cells covering the outermost layer of the fish body (i.e., SECs) do not shed but instead readily divide without undergoing DNA replication, a division mode termed "asynthetic fission" (Chan et al, 2022). Our previous study using *palmskin*, a *brainbow* technology-based tool (Livet et al, 2007; Loulier et al, 2014; Chan et al, 2022) that barcodes SECs in multicolor, showed that the extent and mode of SEC division appear to be influenced by the growth state of the larvae (Chan et al, 2022). In particular, animals with larger size exhibited greater SEC numbers and extents of clonal division. Whereas it is clear that the *palmskin* tool allows for unambiguous real-time observation of diverse cell features–such as cell size, cell number, division orientation, and extent of division–in live individuals of varying sizes (Chan et al, 2022), it has not yet been explored whether these cell attributes are predictive of the animal's ever-changing body size. If the cell-level information can indeed inform body size, it also remains unclear what cell properties best reflect the connection between micrometer-scale features and millimeter-scale readouts.

Machine learning approaches have been increasingly applied in the field of developmental biology to perform various tasks, such as image data segmentation, object recognition, and phenotype classification (Hallou et al, 2021; Naert et al, 2021; Villoutreix, 2021; Greener et al, 2022). As a subfield of machine learning, deep learning models require no manual feature extraction from raw data. The models automatically classify and even recognize high-level features from data with varying levels of abstraction and complexity (Moen et al, 2019). For instance, Čapek et al used bright-field images of normal and signaling pathway-defective zebrafish embryos as training inputs, and found that deep learning

[1]Graduate Institute of Biomedical Electronics and Bioinformatics, National Taiwan University, Taipei, Taiwan [2]Institute of Cellular and Organismic Biology, Academia Sinica, Taipei, Taiwan [3]Department of Electrical Engineering, National Taiwan University, Taipei, Taiwan

Correspondence: acwei86@ntu.edu.tw; chcchen@gate.sinica.edu.tw
*Shang-Ru Yang and Megan Liaw contributed equally to this work

techniques can not only effectively classify different signaling mutants, but the trained models are also able to detect both known and hidden morphological defects in an objective manner (Capek et al, 2023). Although much effort and expertise may be needed to determine appropriate models and optimize hyper parameters for a specific task, deep learning can be a powerful means for linking two seemingly unrelated biological readouts and pinpointing key features that enable such connections (Wu et al, 2019; Capek et al, 2023).

In this study, we captured 722 local, multicolored skin cell images from individual fish, along with their respective whole-body bright-field images, and imputed these data into machine learning models. Using a trained U-Net model (Ronneberger et al, 2015), we determined the exact body surface areas of larval fish from 7 to 12 days post-fertilization (dpf), classifying the fish into three size groups (small [S], medium [M], and large [L]) using K-means clustering. We first applied Cellpose for cell segmentation (Stringer et al, 2021; Stringer & Pachitariu, 2024 *Preprint*) on *palmskin* images, extracting five distinct cell features from each image. Then, we used these features to train a Random Forest machine learning model (Tin Kam, 1995; Breiman, 2001). We found that the model demonstrated modest predictive power at best when considering all features. In contrast, the use of a self-supervised deep learning model to analyze the raw, unsegmented images was much more effective. We determined that a ViT-based model (Dosovitskiy, 2020 *Preprint*), implemented with a random cropping and voting strategy, can readily achieve an F-score of 0.91 in predicting fish size. Intriguingly, systematic trimming of the input image size led us to discover that the anterior–posterior axis contains more growth information than the dorsal–ventral axis, and images of 0.01 mm², or a minimum of 27 cells, are sufficient to reach an F-score of 0.79. By feeding the model with images devoid of cell texture, cell color-encoded information, and cell boundary cues, we further determined that multiplex information, rather than a single attribute, is crucial for the model's prediction accuracy. Finally, we used the unbiased view of Grad-CAM (Selvaraju et al, 2020) to identify that the growth-driven clonal expansion of SECs is one of the key cell features influencing the model's predictions. Taken together, these findings allowed us to identify a deep learning-based pipeline and framework for decoding millimeter-scale organismic information hidden at the cellular level. Our results suggest that the collective signatures of cell dynamics could serve as a real-time proxy for the overall state of an actively growing individual.

## Results

### Multicolored skin cell images and fish sizes as inputs for machine learning

To determine whether the collective features of SECs can be linked to the overall growth state of an individual, we studied a specific post-embryonic growth period of zebrafish larva from 7 to 12 dpf. During this period, SEC images and the animal sizes can be readily captured and quantified (Fig 1A–C). Using the *palmskin* tool, we initiated multicolor barcoding of SECs at 4 dpf and imaged the middle-trunk body region from individual animals of varying sizes between 7 and 12 dpf (two 465 × 465 *µ*m images were captured for each individual, centered on the anal protrusion; the anterior (A) and posterior (P)

regions in Fig S1A). Of note, the zebrafish skin epidermis contains no keratinized dead cell layer, facilitating the high-resolution imaging of SEC populations (Fig 1B). In addition, during this particular growth period, SECs undergo little or no shedding but readily divide on the body surface, leading to variably sized patches of SEC clones (Fig 1D and E). To define the growth state of the animal at this developmental stage, we captured bright-field images from each individual and applied a U-Net based method for automatic segmentation and measurement of the standard length and trunk surface area (see the Materials and Methods section; Fig S1B and C). Of note, both standard length and trunk surface area correlate well with changes in SEC features (Chan et al, 2022). However, we selected trunk surface area as our growth readout as it demonstrates a wider dynamic range during the growth period under study (Fig S1D). To standardize the classification of fish size, we used a K-means clustering algorithm to analyze the measurements of all 361 individuals (across five batches), defining three major size groups: small (S), medium (M), and large (L) (Fig 1F). The 722 SEC images and size labels (i.e., ground truth) for each individual were then used as either training and validation data or testing data for both machine learning and deep learning models (Fig 1G).

### Random forest model demonstrated modest performance for fish size classification

Since *palmskin* images readily reveal several distinct cellular features, we first applied Cellpose to identify cell and clone boundaries. In brief, 231 images were manually segmented using the Cellpose model (Cyto3). The trained model was then applied to segment all remaining images, and a custom Python script was generated to identify and merge neighboring cells of similar color (https://github.com/DANTA-HOJA/AI_SEC/tree/main/script_ml; see the Materials and Methods section for details). We then extracted all of these features from each image, including cell coverage, cell count, cell average size, color patch count, and color patch average size (see the Materials and Methods section; Fig 2A–D). Next, we trained a Random Forest machine learning model on each feature individually and examined the performance. In brief, we allocated 80% of the total images (578 images) and the corresponding size labels for training, although the remaining 20% were reserved for testing. Since differences in sample size will cause class imbalance, we adjusted for the number of images in each class and normalized the prediction value in the confusion matrix to compute a weighted F-score for multi-class comparison (see the Materials and Methods section). The data in Fig 2D show that that learning from each individual cell feature alone resulted in mediocre prediction scores, ranging from 0.42 to 0.66. We then examined whether combining all extracted features could improve prediction accuracy and found that this strategy could improve the F-score to 0.76 (Fig 2E). In an alternative approach, we applied Principal Component Analysis (PCA) to extract image features for machine learning. Of note, PCA is an unsupervised, linear dimensionality-reduction technique commonly used to transform large image datasets for streamlined analysis (Abdi & Williams, 2010). We found that the top five principal components captured the majority of the variability in our imaging data (Fig S2A). However, training with the five principal components did not improve the prediction (F-score = 0.57;

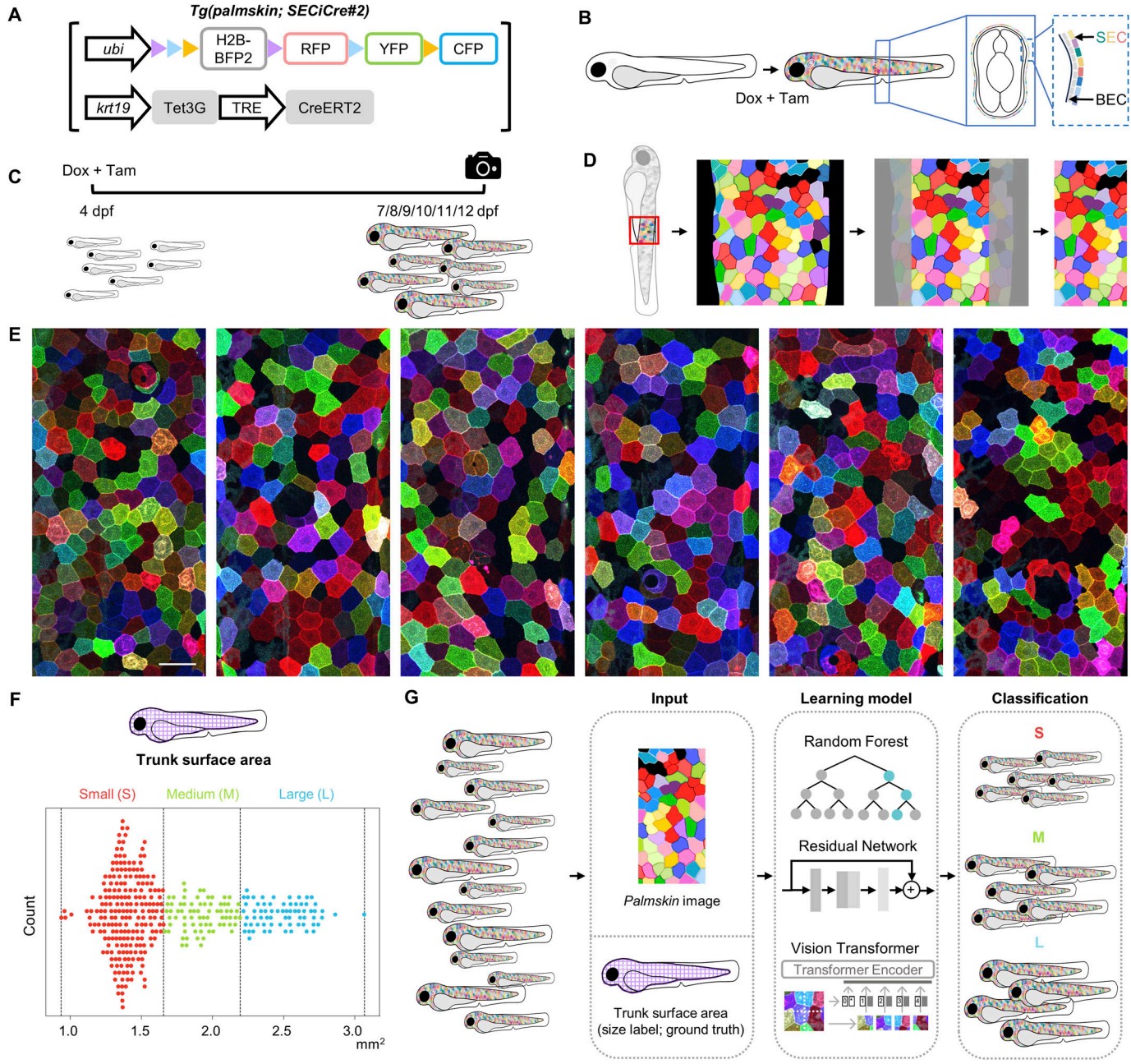

**Figure 1. Multicolor skin cell images captured from live zebrafish larvae as input for machine learning.**
**(A)** The *palmskin* transgenic constructs. **(B)** Schematic cross-section of a *palmskin* larva. Upon a 2-h pulse treatment of tamoxifen (Tam) and doxycycline (Dox), the superficial epithelial cells covering the trunk surface area in the bi-layered epidermis are barcoded in multicolor. SEC, superficial epithelial cell. BEC, basal epithelial cell. **(C)** Schematic timeline illustrating the Dox and Tam treatment protocol and the imaging schedule. dpf, days post-fertilization. **(D)** Schematic illustration of the local *palmskin* images used for machine learning. **(E)** Representative *palmskin* images captured from fish of varying growth sizes. Scale bar, 50 µm. **(F)** K-means clustering of the trunk surface area of 361 fish. Each dot represents an individual fish, with small-sized (S) fish shown in red, medium-sized (M) fish shown in green, and large-sized (L) fish shown in blue. Dotted lines indicate the cut-offs for different size categories. **(G)** Schematic flowchart of the inputs, machine learning models, and growth size classification outputs.

Fig S2B). UMAP analysis based on the five components further showed that the dimensionality-reduction tool failed to cluster fish of the same size category (Fig S2C). Altogether, these findings led us to conclude that the random forest model trained on either defined or objective features of whole *palmskin* images can outperform "random guessing" for prediction of fish size (F-score of 0.76 versus 0.37; Figs 2E and S2D). Thus, local cell images might offer clues about the overall size of the animal. Nevertheless, the suboptimal F-scores suggest that there remains potential for improvement of prediction accuracy.

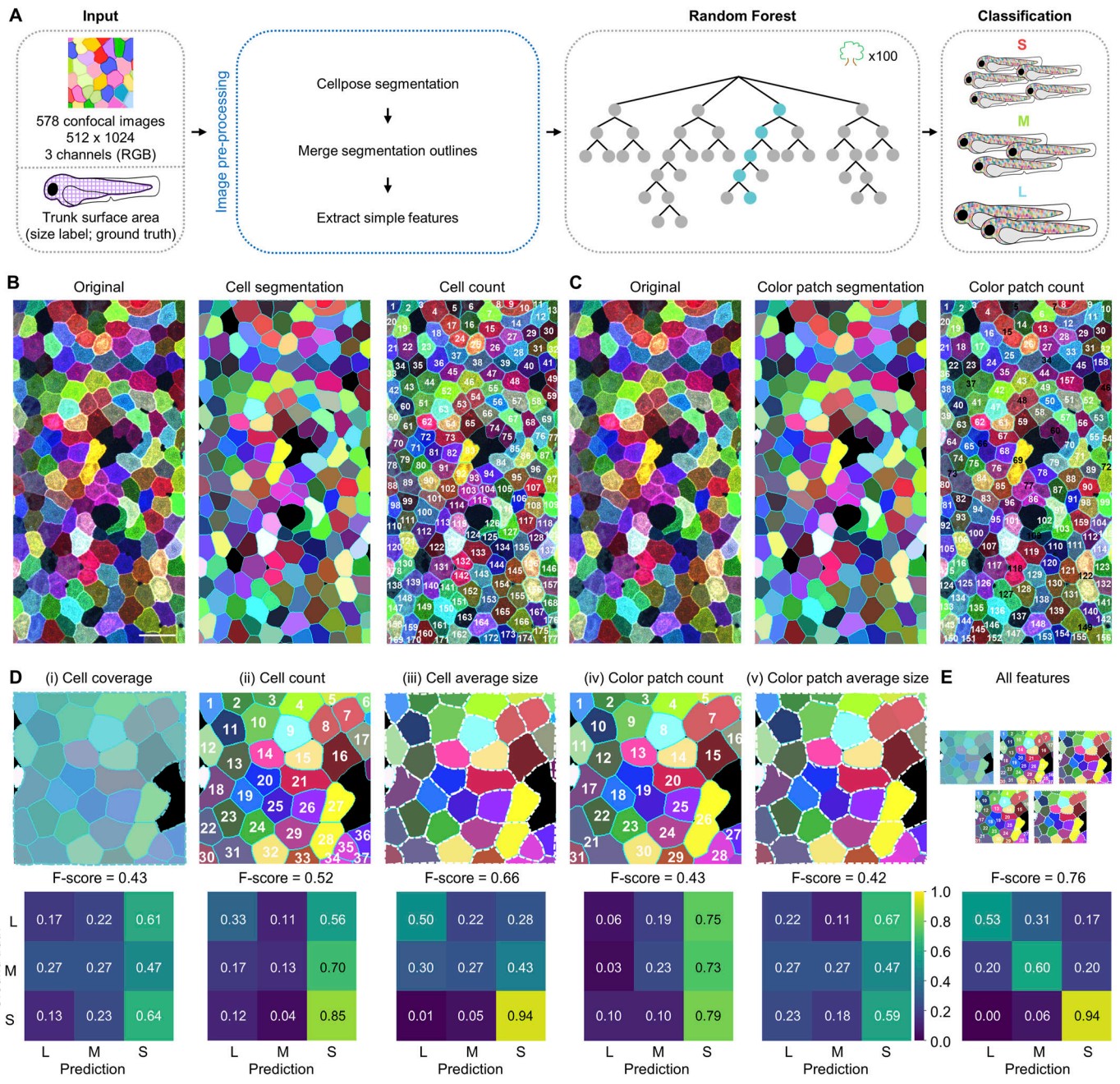

**Figure 2. Random Forest model, trained on defined cellular features, exhibits moderate performance in fish size classification.**
**(A)** Schematic flowchart of the inputs, Random Forest model, and growth size classification output. **(B)** Cell segmentation of *palmskin* images and automated counting of individual SEC cells. The cell numbers are shown in white. Scale bar, 50 μm. **(C)** Color patch segmentation of *palmskin* images and automated counting of individual color patches. The color patch numbers are shown either in white or in black, depending on whether the cells merge with neighboring cells of the same color. **(B, C)** Of note, the same *palmskin* image is shown in both (B, C) for side-by-side comparison. **(D, E)** (Top) Schematic illustrations of the extracted features used for machine learning. (i) "Cell coverage" represents the trunk surface area covered by SEC cells. (ii) "Cell count" represents the counting of individually segmented SEC cells. (iii) "Cell average size" represents the average size of each cell, highlighted by the dotted white borders. (iv) "Color patch count" represents the counting of segmented SEC clones. (v) "Color patch count average size" represents the average size of each clone highlighted by the dotted white borders. (Bottom) Confusion matrix and F-scores of the corresponding extracted features listed above. Scores above 0.7 are shown in black for ease of visualization.

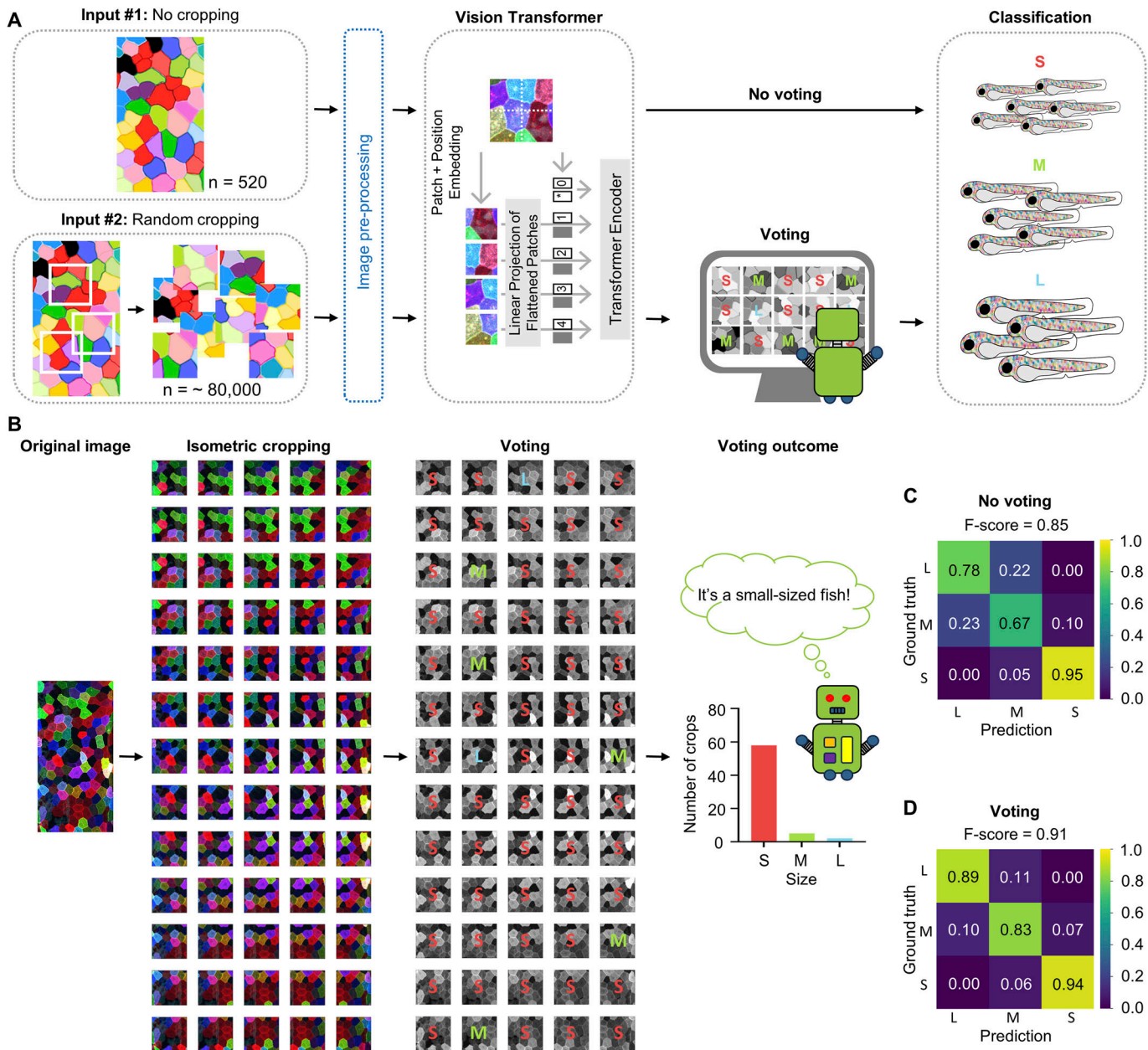

**Figure 3. The Vision Transformer model, with a random crop and voting strategy, achieves an F-score of 0.91.**
**(A)** Schematic flowchart of the inputs, Vision Transformer model, and growth size classification. Two different inputs were fed to the model. Input #1 was the original *palmskin* images; Input #2 had random cropping, a data augmentation technique, applied to the *palmskin* images to generate a larger set of imaging data for the model. **(B)** Schematic flowchart of the isometric cropping and voting. Isometric cropping was used in the test dataset, and voting was used to consolidate all size predictions based on each sub-crop. **(C, D)** Confusion matrix and F-scores for the "No voting" and "Voting" conditions. Scores above 0.7 are shown in black for ease of visualization.

## Vision transformer model with a random cropping and voting strategy boosts F-scores

As *palmskin* images may contain hidden information that reflects the animal's growth state, we next explored whether deep learning models might better predict fish size, as these models can self-learn from raw data without the need for extraction of specific features. To test the idea, we started with the deep learning model Vision Transformer (ViT). In contrast to the

Residual Neural Network (ResNet) and other Convolutional Neural Network-based models that apply convolutional kernels to images, ViT models divide each training image into a series of fixed-size patches and assign a positional embedding to each. Thus, the models are able to incorporate both global and local information from the input images (He et al, 2016; Dosovitskiy, 2020 *Preprint*). Specifically, we used the ViT-B/16 model with pre-trained weights (Torchvision; see the Materials and Methods section for details) for training (Fig 3A). Since the *palmskin*-revealed SEC features

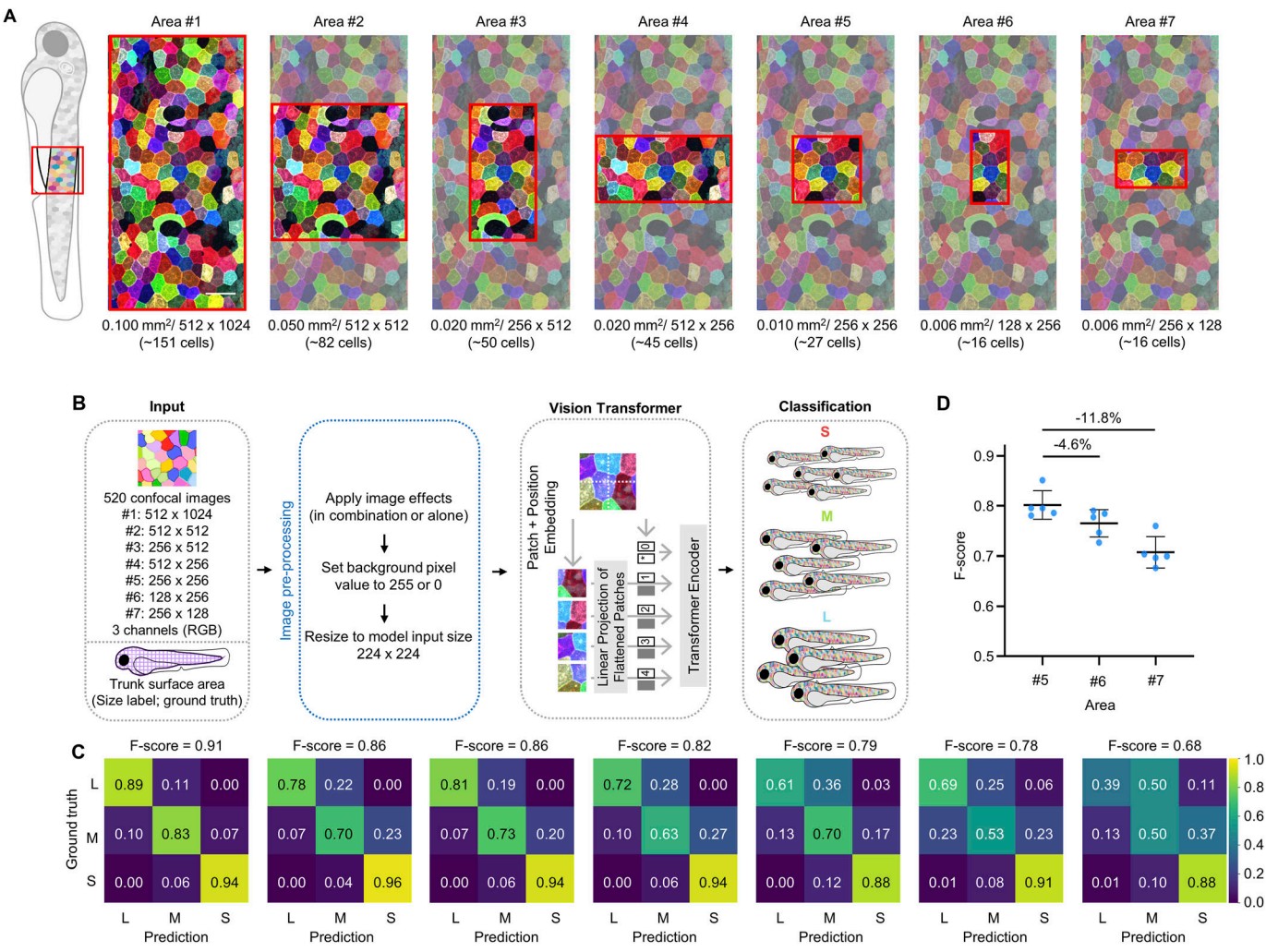

**Figure 4. A single image of 27 skin cells contains information about an individual's overall body size.**
**(A)** Panel of *palmskin* images indicating the different crop sizes used as training inputs. The red boxes highlight the local trunk surface areas of a *palmskin* larva used for training. Below each cropped area, the size of the trunk surface area, pixel size, and the approximate number of cells contained within the boxed area are labeled. Scale bar, 50 μm. **(B)** Schematic flowchart of the inputs, Vision Transformer model, and growth size classification. **(A, C)** Confusion matrix and F-scores for the corresponding crop sizes shown in (A). Scores above 0.7 are shown in black for ease of visualization. **(D)** The F-scores of Areas #5, #6, and #7. Percent differences are shown above the horizontal lines for intergroup comparisons. Five iterations were conducted, and the data are presented as mean ± s.d.

appeared to exhibit regional heterogenicity, we applied a "random cropping" data augmentation technique to the original images, generating additional imaging data for the model to learn from (Krizhevsky et al, 2017). In each training epoch, a random location was selected for cropping, resulting in ~80,000 sub-crops over 167 epochs (Fig S3A; see the Materials and Methods section for details). Correspondingly, each validation and testing image was cropped into a panel of fixed-size and fixed position patches to enable a simple majority vote for the classification (Fig 3B). Intriguingly, we found when using the same input and testing dataset as that used in the Random Forest model (Fig 2), the ViT model exhibited a markedly increased the F-score from 0.76 to 0.85 (Fig 3C). Perhaps unsurprisingly, integrating the random cropping and voting strategy further boosted the overall prediction accuracy to 0.91 (Fig 3D).

To determine the effectiveness of other deep learning models for fish size classification, we replaced the ViT training model with ResNet-50. The same hyperparameters, optimizer, and loss function settings were maintained (Fig S3B). Despite the drastic differences in model architectures, we found that ResNet-50 performed just as well as the ViT model (F-score = 0.85; Fig S3C); moreover, the random cropping and voting strategy similarly enhanced the prediction accuracy to 0.87 (Fig S3D). Taken altogether, these findings led us to conclude that deep learning models trained on raw images consistently outperform machine learning models trained on extracted image features in predicting fish size, as evidenced by the higher F-scores achieved with deep learning models (0.91 and 0.87 versus 0.76 and 0.57; Figs 2, 3, S2, and S3). In addition, straightforward data augmentation techniques, such as random cropping and voting, could readily boost the overall performance of deep learning model analysis of cell images.

## A single image of 27 skin cells readily reveals an animal's dynamic growth state

Given that the deep learning model can achieve high accuracy in predicting fish size based on local cell images, we wanted to determine the minimal number of cells needed in a single *palmskin* image for the model to learn and make reliable predictions. To address the question, we systematically trimmed down the input image sizes–ranging from 0.1, 0.05, 0.02, 0.01, to 0.006 mm$^2$, corresponding to an average of 151, 82, 50, 45, 27, and 16 SECs per image–prior to feeding them into the same model architecture (Fig 4A and B). As expected, we found that prediction accuracy gradually decreased with reductions in image size (and number of imaged cells) (Fig 4C). Intriguingly, the F-scores dropped more sharply when test images were trimmed along the x-axis compared with the y-axis, despite both images being the same size and containing the same number of cells. Specifically, the decrease was 11.8% in Area #7 compared with 4.6% in Area #6, both relative to Area #5 (each containing 16 cells; n = 5 iterations; Fig 4D). These findings suggest that *palmskin* images contain embedded animal growth information with a spatial bias, as the anterior–posterior axis appears to contain more sizing information than the dorsal–ventral axis. Notably, this finding is consistent with the observation that SECs display a biased division orientation along the dorsal–ventral axis at this developmental stage (Chan et al, 2022). Overall, the systematic reduction of input image size showed that a *palmskin* image area of 0.01 mm$^2$, equivalent to 27 cells, is sufficient for the model to make a fair prediction of animal growth size (Area #5 in Fig 4A and C; F-score = 0.79).

To determine which specific cellular features might be crucial for the model's prediction accuracy, we generated an additional set of feature-subtracted images for deep learning and examined the corresponding F-scores. In the "Random color #1" group, SEC clonal information was preserved, but cell texture was removed. In the "Random color #2" groups, both clonal and cell texture information were eliminated. Finally, in the "cell-less" group, cell–cell boundaries were fully removed, leaving only the information related to SEC coverage (Fig S4A and B). Intriguingly, we found that each individual SEC feature contributed to the F-score, even including the basic information of cell coverage or the level of skin cell shedding (i.e., the "cell-less" group in Fig S4C). This finding suggests that each *palmskin* image provides multiplex cellular information, rather than exhibiting a single determinant attribute, and this multiplex information enables the model to accurately predict fish size. Thus, we conclude that despite the distinct length scales of the two readouts, a cell-resolution multicolored image captured from a live animal's body surface (containing as few as two dozen cells) could provide a real-time indication of the animal's overall dynamic growth state.

## Grad-CAM recognizes key cellular features encoded in *palmskin* images

Since multiplex information is crucial for the model's overall performance (Fig S4), we wondered which specific cell features the model detects and relies on most in different size groups. To investigate this question, we applied Grad-CAM to pinpoint the regions within a *palmskin* image that are most essential to the model's decision-making process. Grad-CAM is a visualization technique that generates a coarse heatmap by projecting the weights or probabilities of the trained classes onto the original images, thereby highlighting the key regions influencing the prediction (Selvaraju et al, 2020). Given the model's high accuracy in classifying small and large fish (size-specific F-scores of 0.94 and 0.89, respectively; Fig 3D), we focused the Grad-CAM analysis on these two categories. In brief, we used "pytorch-grad-cam" to generate CAM images by selecting the layer preceding the classifier in the final block of the model encoder. For enhanced visualization, we applied the "COLORMAP_JET" color map from OpenCV to convert the grayscale CAM images to color (Fig 5A). Intriguingly, the small-sized category exhibited Grad-CAM signals that highlighted distinct, isolated patches corresponding to individual SEC cells. In contrast, the signals in the large-sized category were more diffuse, covering broader areas that overlapped with SEC clones. The highlighted regions in the large size group were about 17% larger than those in the small size group (Fig 5B–D). Thus, our findings led us to conclude that the SEC clonal dynamics are likely a key image feature influencing the model's decisions across different size categories. Specifically, the model appeared to rely on the absence of SEC clones in small-sized fish and the presence of the clones in large-sized fish.

To determine whether the deep learning model can classify fish sizes based on gross appearance, we used bright-field images of the fish and the corresponding size labels as training inputs (Fig 5E). Of note, the images were rescaled to a uniform size of 1,288 × 214 pixels to remove size-related information. To our surprise, the same model architecture was able to achieve an F-score of 0.71 for prediction of fish sizes using only bright-field images containing no cellular information. Notably, the model demonstrated particularly high accuracy for the small-sized category (0.97; Fig 5F). However, Grad-CAM analysis later suggested that the model likely makes its best guess based on the relative blank space present in the images, rather than recognizing specific body features (Fig 5G). It is worth noting that we omitted Grad-CAM analysis for the large-sized category because its prediction accuracy was close to random guessing (0.33 versus 0.37; Figs 5F and S2D). Altogether, this initially unexpected finding highlights the importance of retrospective mapping and cross-examination to understand what a deep learning model truly perceives from image data.

## Discussion

In this study, we applied deep learning approaches to bridge two seemingly unrelated biological readouts of cell behavior and animal growth state, which are observed at drastically different length scales. Our results demonstrated that single-cell resolution images of 0.01 mm$^2$, containing an average of 27 cells or 0.6% of the trunk surface area, can embed sufficient information for deep learning models to decode millimeter-scale animal size. Using an objective technique to detect the cellular regions most critical to the model's decisions, we pinpointed the extent and dynamics of skin cell clonal division as key instructive cues for the model's prediction of

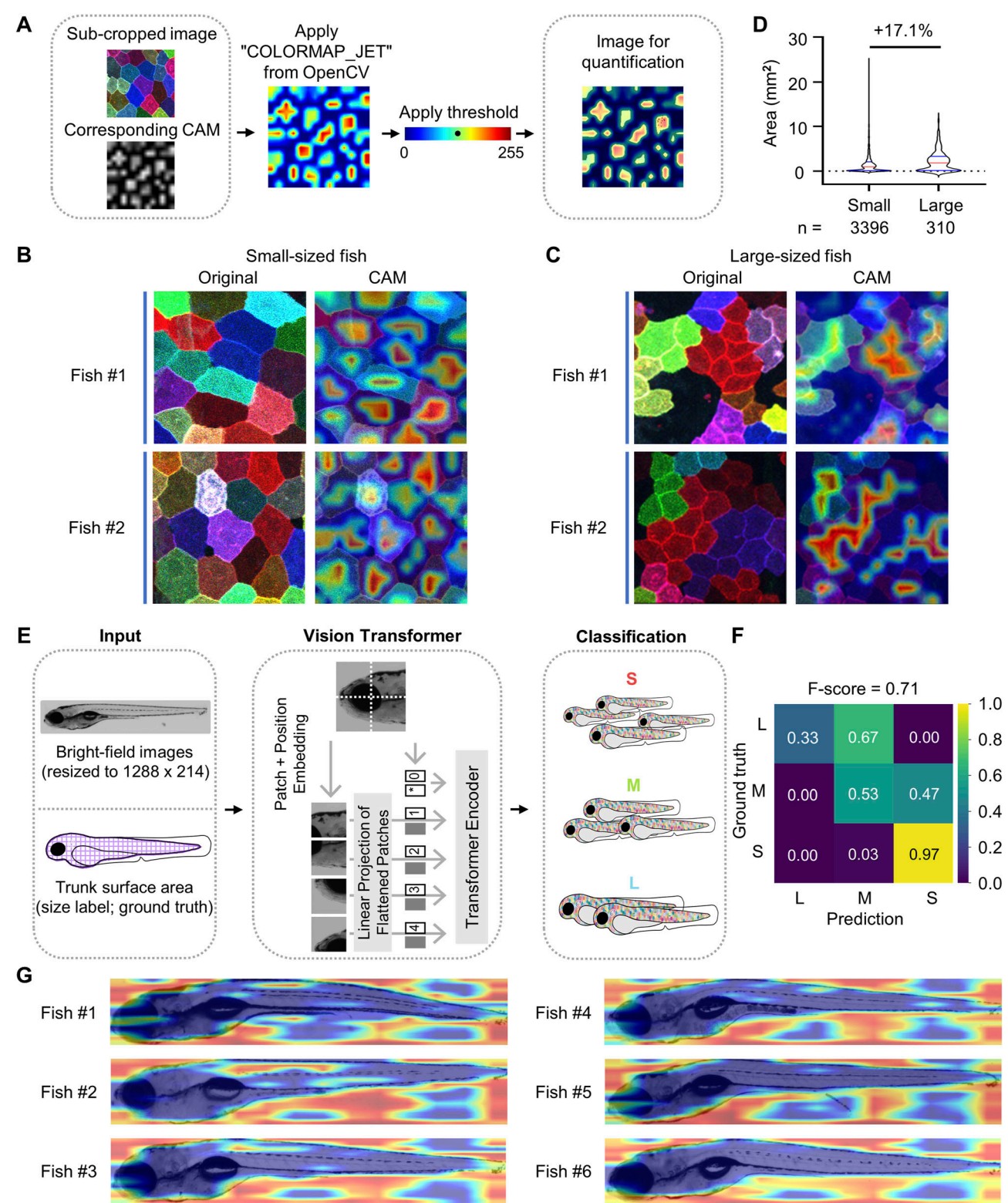

**Figure 5. Grad-CAM recognizes key cellular features important for the model's decisions.**
**(A)** Schematic flowchart of the generation of CAM images for quantification. **(B, C)** Representative *palmskin* images and corresponding CAM images. Two individuals from the small-sized and large-sized categories are shown as indicated. **(D)** Violin plot of all the largest hotspots from each sub-cropped CAM image captured from small- and large-sized fish (red line, median; blue line, quartiles; dotted black line, mean). Percent differences are shown above the horizontal lines for intergroup comparisons. *n* = number of hotspots examined. **(E)** Schematic flowchart of the inputs, Vision Transformer model, and growth size classification. Of note, instead of *palmskin* images, bright-field images and trunk surface area were used as inputs. The input bright-field images were rescaled to a uniform size of 1,288 × 214 pixels to remove size-related

overall animal size. Together, these findings led us to conclude that a snapshot of collective cell behavior at a local scale may serve as a real-time proxy for capturing the dynamic nature of the whole animal growth state. Furthermore, our study reveals a streamlined deep learning-assisted approach for uncovering intricate cross-length scale interactions between cells and the entire organism in a vertebrate model.

Based on the results from deep learning models, we propose that a robust link exists between local skin cell behavior and global animal growth. However, it is yet to be determined what body-wide coupling mechanisms underlie the association. Since the extent of SEC clonal division is positively correlated with local tension levels (Chan et al, 2022), we suspect that dynamic synchronized changes in mechanical forces across the entire organism may act as a real-time coordinator of collective cell responses throughout the animal body. Consistent with this notion, mechanical forces have been shown to mediate organ-wide changes in cell shape and fate, facilitating the formation of hair follicle architecture in developing mouse skin (Villeneuve et al, 2024). Similarly, these forces can also steer long-distance persistent tissue flows to regulate the self-organized growth in quail embryos (Caldarelli et al, 2024). Thus, we speculate that collective information on mechanical force-sensitive cellular targets in living individuals may serve as an indirect yet reliable reporter for certain gross phenotypic changes, such as overall growth state and disease progression.

Whereas the *palmskin* tool can reveal the extent and dynamics of SEC clonal division in live animals, it is important to note that the *palmskin*-labeled multicolored skin cells are a transient population (Chan et al, 2022). The labeled cells are all eventually shed from the animal's body surfaces at later developmental stages, while the newly arising skin cells are label-free and cannot provide further growth information. Moreover, SEC clonal division occurs through an atypical mode of cell division devoid of DNA replication (i.e., asynthetic fission), which takes place predominantly during a specific post-embryonic growth period (Chan et al, 2022). It remains unknown whether this division mode is restricted to skin cells and/or the zebrafish model. Thus, we do not intend to conclude that the tools and trained models are generally applicable to other cell types or for predicting the growth state of other animals. Rather, our findings suggest the existence of whole-body-scale coupling mechanisms connecting microscopic and macroscopic biological readouts. Similar coupling mechanisms may be present in other cellular contexts and whole-animal physiological conditions, and these cross-scale associations could potentially be unraveled through deep learning approaches, even using a relatively small dataset.

Our findings indicate that implementation of a random cropping and voting strategy can markedly enhance the performance of deep learning models in predicting fish sizes from *palmskin* images. However, it is yet to be determined whether this simple data augmentation technique is particularly effective for small image datasets or for cell images with specific characteristics. In the case of *palmskin* images, the extent of SEC division at a population scale is positively correlated with fish growth state. While each SEC exists as a separate unit that may or may not divide, the process of division can be spatiotemporally influenced by neighbors and microenvironments. For instance, recently divided neighbors may alleviate local tension levels, thereby reducing the immediate likelihood of division in adjacent cells. Thus, we suspect that the voting part of the analytic strategy is particularly well-suited for cell images that exhibit extreme spatial heterogeneity while still maintaining intrinsic global coordination. It would be interesting to determine whether the strategy becomes less effective when the information embedded in the input data are less globally coupled. Further work will be needed to assess the applicability and effectiveness of data augmentation techniques on a diverse array of multiplex confocal cell image datasets captured from live animals. Of note, these datasets often require extensive resources to generate and expand, making them less accessible for machine learning/deep learning approaches.

While we classified fish into three different size categories for simplicity in training, we recognize that fish growth is not discrete but a continuous process. This fact may explain the relatively lower F-scores observed for the middle-sized group (Figs 2D and 3D). The rapid development of generative models, such as GANs (Generative Adversarial Networks) and VAEs (Variational Autoencoders) (Goodfellow et al, 2014; Kingma & Welling, 2019), might improve the identification of subtle cellular changes associated with fish of varying sizes, further enabling precise predictions of exact body size. For instance, VAEs can transform the high-dimensional *palmskin* images into a lower-dimensional latent space, preserving or distilling the most essential features needed to reconstruct the images from the latent space. Computer simulations may then be implemented to synthesize cell images that best represent fish of various sizes. This approach could potentially reveal incremental collective features that are instructive of fish growth. The approach might even be suitable to generate de novo images for training and predicting fish that are currently outside the sampled ranges.

In summary, we used deep learning of cell-resolution images captured from live animals to identify specific local cellular features that encode the global state of the animal. We further provide the image datasets, pipelines, codes, and strategies for exploring these intricate cell-organism connections across different length scales. Although our results are highly context dependent, our findings suggest that it is possible to predict the macroscopic form of a living individual from limited microscopic cellular information. Thus, our study highlights the potential of deep learning approaches to accomplish seemingly impracticable tasks.

# Materials and Methods

### Zebrafish lines and maintenance

Zebrafish larvae of 7–12 dpf were imaged in this study. At this developmental stage, animal sex cannot be specified. All larvae

---

information. **(F)** Confusion matrix and F-scores of the training results. Scores above 0.7 are shown in black for ease of visualization. **(G)** Representative CAM images overlaid on the bright-field images. Six individuals from the small-sized category are shown.

were fed with paramecium starting from 8 dpf. The larvae were kept at either a low density (1 larva/100 ml) or a high density (1 larva/10 ml) to maximize the variation in growth sizes. The *Tg(ubi:Palmbow)* [as61] line, also called *palmskin*, and the *Tg(SEC:iCre#2)* [as63] line have been described previously (Chan et al, 2022). All fish lines were maintained at 26°C. Animal experiments were approved by the Institutional Animal Care and Utilization Committee (IACUC) of Academia Sinica. The care and handling of zebrafish were conducted in accordance with IACUC guidelines and regulations.

### Cre activation in *palmskin*

Cre recombination in *Tg(palmskin; SEC:iCre2)* larvae were transiently induced at 4 dpf by a 2-h treatment with tamoxifen (2 $\mu$M; T5648; Sigma-Aldrich) and doxycycline (20 $\mu$g/ml; D9891; Sigma-Aldrich).

### Multicolor live cell imaging in *palmskin*

Animals were anesthetized with tricaine (0.4 mg/ml) and mounted in 1.5% low-melting agarose for imaging. Bright-field images for trunk surface area and standard length measurements of each larva were captured using an M205 stereomicroscope (Leica). All SEC *palmskin* images were captured using an upright SP8 confocal microscope (Leica) equipped with a 25x water-immersion lens (25x/0.95 NA HCXIRAPO). Three lasers were used for excitation: 448, 514, and 552 nm. Image resolution was set at 1,024 × 1,024 pixels, and the image was taken in bidirectional mode. Other imaging settings include: scanning speed of 600 Hz, zoom factor of 1, line average of 2, pinhole size of 2 AU and Z-stack of 2-$\mu$m optical sections. To ensure full coverage of the SEC layer on one side of the fish body, z-stacks spanning 48–84 $\mu$m (~24–42 slides) were captured for each image.

### Image processing

For the automatic measurement of trunk surface area, 248 bright-field images of entire larvae were manually analyzed using ImageJ. The corresponding masks were saved to train a U-Net-based model for segmenting the remaining 113 images. After training, ImageJ was used to determine the trunk surface area of each sample. A modified version of ResNet-18, Res18-UNet, was used for the segmentation task as described (https://github.com/DANTA-HOJA/AI_SEC/tree/main/script_bfseg). For *palmskin* images, two separate images centered on the anal protrusion were captured for each larva (Fig S1A). Maximum projection was applied to all confocal images, which were then merged into RGB images. For the training inputs, images were cropped to dimensions of 512 × 1,024 pixels.

### Growth size categorization and data allocation

To set the ground truth labels for training, a K-means clustering algorithm (Ikotun et al, 2023) was used to group animals into three size categories: small (218 zebrafish), medium (74 zebrafish), and large (69 zebrafish). A random seed of "2022" was set to ensure reproducibility. Of note, the dataset was divided into training and

test sets in an 8:2 ratio in the Random Forest model. The training set was further split into training and validation sets in a 9:1 ratio in the deep learning models. Since each zebrafish contributed two *palmskin* images, the final data allocation in the deep learning models was as follows: 260 zebrafish (520 images) for training, 29 zebrafish (58 images) for validation, and 72 zebrafish (144 images) for testing.

### Feature extraction from *palmskin* images

#### *Cellpose*
To segment cells in *palmskin* images, 231 images were initially processed using the default Cellpose model (Cyto3) (Stringer et al, 2021; Stringer & Pachitariu, 2024 *Preprint*). The segmented results were manually refined and used to fine-tune Cyto3 to enhance segmentation accuracy. Both segmentation channels 1 and 2 were set to 0. The tuned model was then applied to segment the remaining images. To segment "clones" (neighboring cells of the same color), a color distance threshold of 10 was empirically selected and applied using a custom Python script (https://github.com/DANTA-HOJA/AI_SEC/tree/main/script_ml). The results were then used to calculate cell coverage, cell count, and color patch count as described in the workflow (https://github.com/DANTA-HOJA/AI_SEC/tree/main/script_ml). For "cell coverage," after applying a threshold to merge pixels, any pixel not labeled as 0 was included in the calculation. For "cell count," the Cellpose generated labels were counted as individual cells. For "color patch count," the Python script generated labels were counted as individual color patches. Finally, "cell average size" and "color patch average size" were calculated based on the values described above, using the equations provided below.

$$cell\ average\ size = \frac{cell\ coverage}{cell\ count}$$

$$color\ patch\ average\ size = \frac{cell\ coverage}{color\ patch\ count}$$

#### *PCA*
Each *palmskin* image was resized to dimensions of 224 × 224 × 3, resulting in a total of 150,528 pixels per image. The image was then flattened into a 1D array of length 150,528, treating each pixel as a distinct feature. The image dataset used for PCA had an input shape of (578, 150,528); 578 represents the number of images, and 150,528 represents the features per image. The most important features (axes) were identified using explained variance, a statistical measure of dataset variability. The top five components, accounting for an explained variance of 0.15, were used to train a Random Forest model.

### UMAP analysis

The top 5 components extracted from PCA was used as the input for UMAP analysis (McInnes et al, 2018 *Preprint*). Default hyperparameters were used. A random seed was set to "2022" for

reproducibility (https://github.com/DANTA-HOJA/AI_SEC/tree/main/script_ml).

### Random guessing

To generate a pseudo dataset, the "randint" function from NumPy was used. Pseudo-random integers were generated from 0 to 2, representing three different size classes. The ratio of each class in the test dataset was applied to the pseudo-ground truth. The performance of random guessing was then evaluated using the generated pseudo dataset.

### Random cropping and voting

To expand the sample size and dataset coverage, a random location was selected for cropping during each training epoch, resulting in ~80,000 sub-crops over 167 epochs. In addition, random rotations, flipping, contrast adjustments, Gaussian blur, and sharpening were applied using the "imgaug" library (https://github.com/aleju/imgaug) to enhance the model's generalization capability. The applied augmentations were executed in a top-down manner, with specific probabilities assigned to each transformation. In brief, Fliplr and Flipud were used for horizontal and vertical flipping. GammaContrast and SigmoidContrast were applied as an exclusive pair with a probability of 0.3, ensuring that only one was applied per instance. Similarly, GaussianBlur and Sharpen were used as an exclusive pair with a probability of 0.3. Random rotations were applied within a range of −30° to 30° to enhance generalization only in the voting case. For model evaluation, isometric cropping was applied to the test dataset. Predictions from cropped images of each fish were aggregated using a voting mechanism, where each cropped image contributed one vote. Images lacking cell information were excluded. The final predicted fish size was determined as the class receiving the most votes (https://github.com/DANTA-HOJA/AI_SEC/tree/main/script_dl).

### Generation of feature-subtracted images

Feature-subtracted images were generated using segmentation results obtained from Cellpose after applying the color distance threshold. For "Random color #1" images, each cell or clone was randomly assigned a unique six-digit identifier. This identifier was then converted to a HEX color code using the "hex2rgb" function from the Python package "matplotlib." The HEX color codes generated for each cell or clone were used to assign pseudo-colors, and the resulting images were exported as RGB files. For "Random color #2" images, the segmentation results from Cellpose before applying the color distance threshold were used. Each segmented cell/clone was assigned a six-digit identifier corresponding to the previously generated numbers for "Random color #1." As the color distance threshold was not applied to this set, clones or neighboring cells of the same color were not merged. Remaining cells without the identifier were randomly assigned new identifiers. All numbers were then converted to HEX color codes and exported as RGB files. Finally, for "Cell-less" images, segmentation backgrounds with a value of "0" were set to black, while all values above 0 were set to a 50% gray tone with an intensity value of 127 (https://github.com/DANTA-HOJA/AI_SEC/tree/main/script_adv).

## Model training and size classification

### Random forest

Features extracted from Cellpose or identified through PCA were used as inputs to train a random forest model using the Python package "scikit-learn." The model's default hyperparameters were applied with the number of estimators set to 100. A random seed was set to "2022" for reproducibility (https://github.com/DANTA-HOJA/AI_SEC/tree/main/script_ml).

### Vision Transformer (ViT)

The ViT-B/16 model ("vit_b_16") from "torchvision" with "IMAGE-NET1K_V1" (Deng et al, 2009) pre-trained weights was used. This model comprises 12 transformer encoder blocks and processes input images resized to 224 × 224 pixels. The "16" indicates the patch size, meaning the input image was divided into 16 × 16-pixel patches, resulting in a total of 196 patches ([224/16] × [224/16]). Each patch had three channels corresponding to the standard RGB image format, resulting in a (16 × 16 × 3) or 768-dimensional representation. Thus, each 16 × 16 patch was represented as a 768-dimensional vector. A sequence of 196 such vectors was formed and then processed by the transformer model. During training, the transformer weights were fine-tuned using the AdamW optimizer for rapid convergence (Loshchilov & Hutter, 2017 Preprint) with a weight decay of $1 \times 10^{-2}$. The learning rate was set to $1 \times 10^{-5}$. Training was conducted for a maximum of 500 epochs with an early stopping criterion of 50 epochs. The batch size was set to 64. A random seed of "2022" was used to ensure reproducibility. Softmax Cross-Entropy was used as the loss function to penalize poor predictions (https://github.com/DANTA-HOJA/AI_SEC/tree/main/script_dl). For model training, three different input types were used with the same model: (1) Cropped Images: images of varying sizes were cropped to areas of 0.100, 0.050, 0.020, 0.010, and 0.006 mm$^2$. After cropping, the images underwent the same random cropping and voting strategy as described above. Random seeds were respectively set to "2022," "2023," "2024," "2025," and "2026" (for different cropping sizes) to ensure reproducibility. (2) Feature-Subtracted Images: the images were processed using the same random cropping and voting strategy. The cropping area was fixed at 0.01 mm$^2$. (3) Bright-Field Images: the images were processed using the same method but without the random cropping and voting step. For pre-processing, a U-Net-generated segmentation mask was used to determine the appropriate resizing scale for the entire dataset. Bounding boxes for individual segmentation masks were automatically generated using the "boundingRect" function from OpenCV. The mean size of these bounding boxes was calculated and used as a reference standard; all images were resized to 1,288 × 214 pixels.

### Residual Network (ResNet)

The ResNet-50 residual network architecture from "torchvision" was used. Consistent with the ViT model, the "IMAGENET1K_V1" pre-trained weights were used. The model consists of 50 layers, with input image dimensions set to 224 × 224 pixels. Each input image includes three channels corresponding to the standard RGB format. During training, the backpropagation algorithm was used to compute gradient updates, and the optimizer adjusted the model weights accordingly. The model was fine-tuned using the AdamW optimizer, consistent with the ViT-B/16 model (https://github.com/DANTA-HOJA/AI_SEC/tree/main/script_dl).

To prevent both deep learning models from learning background regions without cells, the input image backgrounds were modified during training. The RGB images were converted to the HSB (hue, saturation, brightness) color model, and the pixel values in the background were set to both the maximum (255) and minimum (0) values for an eight-bit image, thereby generating two images that differed only in their backgrounds. The background was then determined based on pixel values in the brightness channel of the HSB model. These two modified images were then fed into the model to obtain their respective output logits, where "logits" refer to the model's raw numerical outputs. Since the prediction involves three classes (small, medium, and large), each set of logits was represented as a 1 × 3 matrix, corresponding to the model's confidence levels for each class. If the logits of the two input images were similar, it was concluded that the model perceived the two images with different backgrounds as comparable and did not rely on background information to make predictions. To evaluate the difference between the logits, the mean squared error (MSE) was calculated. A low MSE value indicates that the model perceived the two versions of the image as nearly identical, meaning it had ignored background differences. A high MSE value suggested that the model was still influenced by background variations. Both MSE loss (which promotes background invariance) and cross-entropy (CE) loss (which ensures correct classification) were used together to guide the AdamW optimizer in adjusting the model's internal weights (https://github.com/DANTA-HOJA/AI_SEC/tree/main/script_dl).

$$MSE = \frac{1}{N}\sum_{i=1}^{N}\left(y_i - \widehat{y_i}\right)^2,$$

where $N$ is the number of samples, $y_i$ is the true value of sample $i$, and $\widehat{y_i}$ is the predicted value of sample $i$.

$$CE = -\sum_{i=1}^{N}\sum_{c=1}^{C} y_{i,c}\log\left(\widehat{y_{i,c}}\right),$$

where $N$ is the number of samples and $C$ is the number of classes. $y_{i,c}$ is the true label of sample $i$ for class $c$ (usually in one-hot encoding, where one class is 1 and the rest are 0). $\widehat{y_{i,c}}$ is the predicted probability of sample $i$ belonging to class $c$ (e.g., the output of a softmax function).

In addition, an intensity threshold of 30 was applied to the brightness channel, so pixels with values equal to or below 30 (on a scale of 0–255 for an eight-bit image) were considered dark and part of the background. A threshold was also established to automatically exclude sub-cropped images from the validation and test sets, referred to as the dark ratio. In these analyses, the dark ratio was set to 0.65, i.e., if 65% or more of the pixels in an image were dark, the image was excluded from the validation and test sets (https://github.com/DANTA-HOJA/AI_SEC/tree/main/script_dl).

$$dark\_ratio = \frac{\# \ of \ pixels \ under \ intensity \ threshold}{total \ \# \ of \ pixels \ in \ the \ image}$$

For both model trainings, a batch size of n = 64 was used, and training was conducted on an NVIDIA RTX 4090 GPU with 24GB VRAM (https://github.com/DANTA-HOJA/AI_SEC).

## Evaluation of classification efficiency

The F1 score was used as the primary performance metric to evaluate model effectiveness.

$$F1\,score = 2 \times \frac{precision \times recall}{precision + recall}$$

The F1 score is a balanced metric that evaluates both precision and recall. Precision represents the proportion of correctly predicted positive cases (true positive) out of all predicted positives (true positive and false positive). Recall represents the proportion of true positives out of all actual positives (true positive and false negative). The best model in this study was selected based on the highest validation score. Given that datasets may expand to have more samples across classes, the highest validation score was considered as an average of two F1 metrics: macro F1 score and weighted F1 score. The macro F1 score was computed as the average F1 score across all three classes. It treats each class independently, regardless of sample size. The weighted F1 score accounts for class imbalances by weighting the F1 scores based on the number of cases in each class.

$$Weighted\ F1 = \frac{\sum_{i=1}^{C} w_i \cdot F1_i}{\sum_{i=1}^{C} w_i} \ (or\ F\text{-}score)$$

$$Macro\ F1 = \frac{1}{C}\sum_{i=1}^{C} F1_i$$

$$F1\,score\,for\,validation = (Weighted\ F1 + Macro\ F1)/2$$

where $C$ is total number of classes (for this task, $C = 3$), $w_i$ is the weight of class $i$, typically the sample size $n_i$ of class $i$, so $w_i = n_i$, and $F1_i = \frac{2 \cdot Precision_i \cdot Recall_i}{Precision_i + Recall_i}$.

However, to account for potential batch effects and data imbalance in the overall samples, the final performance metric was reported as the weighted F-score. This parameter is referred to as the "F-score" throughout the text.

## Gradient-weighted class activation mapping (Grad-CAM)

Grad-CAM from "Pytorch-grad-cam" was used to visualize the importance of features recognized by the deep learning models. In brief, the layer preceding the classifier in the final block of the model's encoder was selected. This layer contains high-level feature representations and is therefore suitable for identifying regions most influential in the model's predictions. Since the generated CAMs were initially represented in grayscale, the "jet" colormap from "OpenCV" was applied to convert the grayscale

images to color for improved visualization. This color conversion enhanced visualization by mapping low-activation areas to cool colors (blue and green) and high-activation areas to warm colors (red and yellow). All cropped images that underwent voting were then arranged to align with their corresponding positions in the original image. To quantify the model's attention regions across different image sizes, a CAM threshold was set to 120 (close to half of the maximum value of 255 for an eight-bit image) to retain hotspots while isolating them. To focus the activation specifically on the SECs, a mask was created from the original image. The original image was converted to HSB, and a threshold of 30 was applied to the brightness channel to generate a binary mask. A 2 × 2 erosion operation was applied to remove remaining bright spots in the background, followed by a 2 × 2 dilation to refine the mask. Finally, an "AND" operation was performed between the thresholded CAM and the mask. Quantification of the attention regions was performed by calculating the value of the largest hotspot in each cropped S-size and L-size CAM image (https://github.com/DANTA-HOJA/AI_SEC/tree/main/script_dl).

## Data Availability

All image data used in the study are available on a public repository (Yang et al, 2025). All codes and analytical pipelines have been deposited along with detailed step-by-step instructions (https://github.com/DANTA-HOJA/AI_SEC).

## Supplementary Information

## Acknowledgements

We thank Kai-Fan Wu and Yue Rong Tan for providing the segmented *palmskin* files for Cellpose training; Iris Chen and the Taiwan Zebrafish Core Facility at Academia Sinica (TZCAS; NSTC 112-2740-B-400-001) for their maintenance of zebrafish lines; Dr. Sheng-Feng Shen and the members of the Chen laboratory for their feedback on the manuscript, and Marcus J Calkins for English editing and comments. We acknowledge the Leap Fellowship of the Foundation for the Advancement of Outstanding Scholarship, intramural funding from the Institute of Cellular and Organismic Biology, Academia Sinica, to C-H Chen, grant support from Academia Sinica (AS-CDA-109-L03, AS-GCS-112-L01, and AS-IV-114-L01-ASSA), and the National Science and Technology Council (NSTC), Taiwan, to C-H Chen (NSTC 113-2311-B-001-022-MY3), and to A-C Wei (NSTC 113-2221-E-002-048-MY3); National Taiwan University Center for Advanced Computing and Imaging in Biomedicine and National Health Research Institutes to A-C Wei (NTU-112L900701 and NHRI-EX-114-11121SC).

### Author Contributions

S-R Yang: conceptualization, data curation, software, formal analysis, supervision, validation, investigation, visualization, methodology, project administration, and writing—original draft, review, and editing.

M Liaw: data curation, software, formal analysis, validation, investigation, visualization, methodology, and writing—review and editing.

A-C Wei: data curation, formal analysis, supervision, funding acquisition, validation, investigation, visualization, methodology, project administration, and writing—review and editing.

C-H Chen: conceptualization, supervision, funding acquisition, validation, investigation, visualization, methodology, project administration, and writing—original draft, review, and editing.

### Conflict of Interest Statement

The authors declare that they have no conflict of interest.

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
