## [Reviewer comments · Life Science Alliance]

Life Science Alliance

Deep learning models link local cellular features with whole-animal growth dynamics in zebrafish

Shang-Ru Yang, Megan Liaw, An-Chi Wei, and Chen-Hui Chen

DOI: <https://doi.org/10.26508/lsa.202503319>

Corresponding author(s): *Chen-Hui Chen, Institute of Cellular and Organismic Biology, Academia Sinica*

Review Timeline:

Submission Date:	2025-03-20
Editorial Decision:	2025-05-01
Revision Received:	2025-05-08
Accepted:	2025-05-12

Scientific Editor: Tim Fessenden

Transaction Report:

May 1, 2025

RE: Life Science Alliance Manuscript #LSA-2025-03319-T

Dr. Chen-Hui Chen
Institute of Cellular and Organismic Biology, Academia Sinica
128 Sec.2, Academia Rd
Taipei, Taipei 11529
Taiwan

Dear Dr. Chen,

Thank you for submitting your manuscript entitled "Deep learning models link local cellular features with whole-animal growth dynamics in zebrafish". This manuscript was evaluated by two experts in this field, whose comments are included below. As you will see, both reviewers commended the intriguing findings on organismal development along with the deep learning models that will benefit researchers in this field. Reviewers had only minor suggestions which we invite you to consider. We would be happy to publish your paper in Life Science Alliance pending those changes and final revisions necessary to meet our formatting guidelines.

- please be sure that the authorship listing and order is correct.
- Please add ORCID ID for corresponding (and secondary corresponding) author--you should have received instructions on how to do so.
- Please add a Summary Blurb/Alternate Abstract in our system.
- Please add Keywords and a Category for your manuscript in our system.
- Please add the X and Bluesky handles of your host institute/organization, as well as your own and/or one of the authors, to our system.
- The titles in both the system and the manuscript file must be consistent with each other.
- Please remove the "highlights" section from the title page.
- Please add author contributions to our system as well.

LSA now encourages authors to provide a 30-60 second video where the study is briefly explained. We will use these videos on social media to promote the published paper and the presenting author (for examples, see <https://docs.google.com/document/d/1-UWCfbE4pGcDdcgzcmiuJI2XMBJnxKYeqRvLLrLS08s/edit?usp=sharing>). Corresponding or first-authors are welcome to submit the video. Please submit only one video per manuscript. The video can be emailed to contact@life-science-alliance.org

A. FINAL FILES:

B. MANUSCRIPT ORGANIZATION AND FORMATTING:

Sincerely,

Reviewer #2 (Comments to the Authors (Required)):

This manuscript addresses an interesting and important topic in developmental biology and other fields, namely the degree to which features at one scale predict those at an entirely different scale. Indeed, this is an area of critical importance if we are ever to understand how cells and their behaviors are translated into morphology at tissue, organ or organismal level. Here, the use cutting edge analytical approaches to address whether cell-level features acquired from just a small area can predict overall animal size, exploiting the zebrafish skin, previously studied by this group and having well-documented features of morphogenesis. The overall conclusion is that such inferences can indeed be made, but the strengths of classification depend greatly on the particulars of the approach.

I cannot speak to the specifics of the models and how they have been used in this context. If these methods are sound, however, then the paper makes a valuable contribution and is likely to stimulate further work in this area by pointing to approaches only now becoming available and of potentially great impact to the field.

Reviewer #3 (Comments to the Authors (Required)):

In this manuscript, Yang et al. present a timely and relevant „pars pro toto" study that demonstrates the ability of deep learning models to link microscopic cellular features with macroscopic organismal traits. Specifically, the authors employed a Vision Transformer-based approach to predict overall zebrafish larval body size from single-cell-resolution images of superficial epithelial cells. The central conclusion that there is a cross-scale association and tight coupling between microscopic cellular patterns and macroscopic organismal size is exciting and convincingly supported by the presented data. The authors provide a thorough dissection of the parameters that contribute to the information relevant for the classification model, and they highlight intriguing differences along the anterior-posterior and dorsal-ventral axes. Impressively, their deep learning approach remains effective even with simpler bright-field images, which greatly broadens the method's accessibility and practical value. A major strength of the manuscript is the clarity and robustness of the presented methods and results. The figures are particularly well-crafted and effectively illustrate the key findings. The authors carefully phrase their conclusions and explicitly acknowledge and point out the limitations of their methodology and system. Additionally, they provide open access to their datasets, models, and code, which promotes transparency and will facilitate broader adoption by the scientific community.

I have only a few minor suggestions:

- 1) The statement, "Grad-CAM analysis later suggested that the model likely makes its best guess based on the relative blank space present in the images, rather than recognizing specific body features (Figure 5G)" currently lacks sufficient clarity. Adding further detail and including comparisons across all size categories, rather than showing examples exclusively from small-sized individuals, would help clarify this interpretation.
- 2) Please specify the number of slices recorded in the z-stacks.
- 3) For Figures 3A, 4B, 5E, and S4A, I suggest replacing the Institute of Cellular and Organismic Biology gate with actual images to better reflect the experimental conditions.
- 4) In the discussion, please correct the typographical error "truck" to "trunk."
- 5) The abbreviation "F.S." mentioned in the Author contributions section does not correspond to any listed co-author. Please clarify or correct this discrepancy accordingly.

Reviewer #2 (Comments to the Authors (Required)):

This manuscript addresses an interesting and important topic in developmental biology and other fields, namely the degree to which features at one scale predict those at an entirely different scale. Indeed, this is an area of critical importance if we are ever to understand how cells and their behaviors are translated into morphology at tissue, organ or organismal level. Here, the use cutting edge analytical approaches to address whether cell-level features acquired from just a small area can predict overall animal size, exploiting the zebrafish skin, previously studied by this group and having well-documented features of morphogenesis. The overall conclusion is that such inferences can indeed be made, but the strengths of classification depend greatly on the particulars of the approach.

I cannot speak to the specifics of the models and how they have been used in this context. If these methods are sound, however, then the paper makes a valuable contribution and is likely to stimulate further work in this area by pointing to approaches only now becoming available and of potentially great impact to the field.

Response: We thank the reviewer for their enthusiastic comments.

Reviewer #3 (Comments to the Authors (Required)):

In this manuscript, Yang et al. present a timely and relevant „pars pro toto" study that demonstrates the ability of deep learning models to link microscopic cellular features with macroscopic organismal traits. Specifically, the authors employed a Vision Transformer-based approach to predict overall zebrafish larval body size from single-cell-resolution images of superficial epithelial cells. The central conclusion that there is a cross-scale association and tight coupling between microscopic cellular patterns and macroscopic organismal size is exciting and convincingly supported by the presented data. The authors provide a thorough dissection of the parameters that contribute to the information relevant for the classification model, and they highlight intriguing differences along the anterior-posterior and dorsal-ventral axes. Impressively, their deep learning approach remains effective even with simpler bright-field images, which greatly broadens the method's accessibility and practical value. A major strength of the manuscript is the clarity and robustness of the presented methods and results. The figures are particularly well-crafted and effectively illustrate the key findings. The authors carefully phrase their conclusions and explicitly acknowledge and point out the limitations of their methodology and system. Additionally, they provide open access to their datasets, models, and code, which promotes transparency and will facilitate broader adoption by the scientific community.

I have only a few minor suggestions:

- 1) The statement, "Grad-CAM analysis later suggested that the model likely makes its best guess based on the relative blank space present in the images, rather than recognizing specific body

features (Figure 5G)" currently lacks sufficient clarity. Adding further detail and including comparisons across all size categories, rather than showing examples exclusively from small-sized individuals, would help clarify this interpretation.

Response (3.1): We thank the reviewer for the question as it provides us an opportunity to better explain the rationale for presenting examples exclusively from small-sized individuals. While the model demonstrated high accuracy in the small-sized category ($F = 0.97$; Figure 5F), its performance in the large-sized category was close to random guessing (0.33 vs. 0.37; Figure S2D). Thus, applying Grad-CAM analysis to identify key regions leading to 'random guessing' would have little or no interpretive value. To clarify this point, we have revised the text on page 13, paragraph 2:

"However, Grad-CAM analysis later suggested that the model likely makes its best guess based on the relative blank space present in the images, rather than recognizing specific body features (Fig 5G). It is worth noting that we omitted Grad-CAM analysis for the large-sized category because its prediction accuracy was close to random guessing (0.33 vs 0.37; Figs 5F and S2D). Altogether, this initially unexpected finding highlights the importance of retrospective mapping and cross-examination to understand what a deep learning model truly perceives from image data."

2) Please specify the number of slices recorded in the z-stacks.

Response (3.2): As suggested, we have included the number of slices recorded in the z-stacks on page 18, paragraph 3:

"To ensure full coverage of the SEC layer on one side of the fish body, z-stacks spanning 48-84 μm (approximately 24 to 42 slides) were captured for each image."

3) For Figures 3A, 4B, 5E, and S4A, I suggest replacing the Institute of Cellular and Organismic Biology gate with actual images to better reflect the experimental conditions.

Response (3.3): As advised, we have replaced the symbolic photo with actual *palmskin* images in Figures 1G, 3A, 4B, 5E, and S4A for consistency and clarity.

4) In the discussion, please correct the typographical error "truck" to "trunk."

Response (3.4): We have corrected the typo on page 14.

5) The abbreviation "F.S." mentioned in the Author contributions section does not correspond to any listed co-author. Please clarify or correct this discrepancy accordingly.

Response (3.5): We have removed the misplaced abbreviation from the section.

May 12, 2025

RE: Life Science Alliance Manuscript #LSA-2025-03319-TR

Dr. Chen-Hui Chen
Institute of Cellular and Organismic Biology, Academia Sinica
128 Sec.2, Academia Rd
Nankang
Taipei, Taipei 11529
Taiwan

Dear Dr. Chen,

Thank you for submitting your Research Article entitled "Deep learning models link local cellular features with whole-animal growth dynamics in zebrafish". It is a pleasure to let you know that your manuscript is now accepted for publication in Life Science Alliance. Congratulations on this interesting work.

DISTRIBUTION OF MATERIALS:

Again, congratulations on a very nice paper. I hope you found the review process to be constructive and are pleased with how the manuscript was handled editorially. We look forward to future exciting submissions from your lab.

Sincerely,
